



# Soil Aggregate Stability of Forest Islands and Adjacent Ecosystems in West Africa

Amelie Baomalgré Bougma[1] [*], Korodjouma Ouattara[1], Halidou Compaore[1], Hassan Bismarck. Nacro[2], Caleb Melenya[3], Samuel Ayodele Mesele[4], Vincent Logah[3], Azeez Jamiu Oladipupo[4], Elmar Veenendaal[5], Jonathan Lloyd[6, 7, 8]

[1] Institut de l'Environnement et de Recherches Agricoles (INERA), Burkina Faso, 04 BP 8645 Ouagadougou 04, Burkina Faso

[2] Université Nazi BONI (UBN), 01 BP 910, Bobo, 01 Burkina Faso.

[3] Kwame Nkrumah University of Science and Technology, Kumasi, Ghana

[4] Federal University of Agriculture, Abeokuta (FUNAAB), PMB 2240 Abeokuta, Nigeria.

[5] Nature Conservation and Plant Ecology Group, Wageningen University and Research, Droevendaalsesteeg 3a, 6700 AA, Wageningen, Netherlands

[6] Department of Life Science, Imperial College London, Silwood Park Campus, Buckhurst Road, Ascot, SL5 7PY, UK.

[7] School of Tropical and Marine Sciences and Centre for Terrestrial Environmental and Sustainability Sciences, James Cook University, Cairns, 4870, Queensland, Australia

[8] Universidade de São Paulo, Faculdade de Filosofia Ciências e Letras de Ribeirão Preto, Av Bandeirantes, 3900 , CEP 14040-901, Bairro Monte Alegre , Ribeirão Preto, SP, Brazil

*Correspondence to*: Amelie B. Bougma (ameliebougma@yahoo.fr )

**Abstract.** In the more mesic savanna areas of West Africa, significant areas of relatively tall and dense vegetation with a species composition more characteristic of forest than savanna are often found around villages areas. These 'forest islands' may be the direct action of human activity. To better understand the processes leading to the development of these patches with relatively luxuriant vegetation, our study focused on the stability of the soil aggregates of forest islands, nearby areas of natural savanna vegetation across a precipitation transect in West Africa for which mean annual precipitation at the study sites ranges from 0.80 to 1.27 m a$^{-1}$. Soil samples were taken from 0 to 5 cm and 5 to 10 cm depths and aggregate fractions with diameters: > 500 μm, 500-250 μm and 250-53 μm (viz. "macro aggregates", "mesoaggregates" and "microaggregates") determined using the water sieving method. The results showed significant higher proportion of stable meso and macro-aggregates in forest islands and natural savanna compared to agricultural soils ($p < 0.05$). On the other hand, although there was no effect of land-use type on microaggregates stabilities, there was a strong tendency for the micro-aggregate fraction across all land use types to increase with increasing precipitation. Simple regression analyses showed soil organic carbon and iron oxides contents as the most important factors influencing aggregate stability in West African ecosystems.

**Keywords:** Sites, land use, macro-aggregates, micro-aggregates, West Africa

## 1. Introduction

In West Africa, both natural and human dominated ecosystems are often affected by land degradation processes, with soil erosion usually considered the most severe threat to long term sustainability. The erosion process itself results from a complex combination of climatic and anthropogenic factors (Zombre, 2003). In general, aggregate stability is a key metric used for





assessing soil susceptibility to erosion (Barthès and Roose, 2002) as it strongly influences  the rates of  water infiltration and
runoff, and  plays a key role in the dynamics and stabilization of soil organic matter (Six et al., 2000). The aggregate formation
process itself is a complex process influenced by soil organic matter content, climate conditions, soil type, soil mineralogy and
land use patterns (Ezebilo, 2004;  Six et al., 2004; Ouattara et al., 2008; Mataix-Solera et al., 2011). Most recently, several
studies showed the role of soil organisms and vegetation structure and/or species composition as additional factors influencing
the stability of soil aggregates (Six et al., 2000; Chartier et al., 2011; Berendse et al., 2015; Gould, 2016). With a species and
structural composition more typical of forest stands found in humid regions, "islands" of dense vegetation typically of 0.1 to
10 ha area are often found surrounding many village areas in the West African mesic savanna zones where they are thought to
have resulted from, at least in part, from the conscious actions of the nearby village occupants (Leach and Fairhead, 1995;
Jones, 1963). There have, however, been few studies on the role of such "Forest Islands" (FI) and their unique ecological
characteristics (Kokou and Sokpon, 2006), apart from the descriptive analyses of few soil profiles ( Sobey, 1978; Fairhead and
Leach, 1998).
This study aims to contribute to the knowledge of the edaphic properties of FI (Forest island) through by assessing soil
aggregate size distributions in adjacent savannas (considered to be the typical 'natural' vegetation of the region) and cultivated
fields. Considering some recent studies on the importance of biodiversity and vegetation cover on soil quality (Chartier et al.,
2011; Berendse et al., 2015; Gould, 2016), we hypothesized  that soil aggregate stability is higher under forest islands than in
adjacent savanna or agricultural field.
**2.  Material and methods**
**2.1 Sampling locations and site descriptions**
The study was carried out in 2016 in 11 locations across Burkina Faso, Ghana and Nigeria. The study sites were distributed
across three agro-ecological zones (AEZ) (Figure 1) as defined by Ker (1995). At each of the eleven location, three land use
types were selected for sampling as follows:
**2.1.1 Forest island (FI)** plots consisted of patches of forests around villages with open landscape mosaic of relatively open
savanna vegetation and agricultural fields. The trees are tall, being 15 to 20 m high with typically more than 400 individuals
per hectare with diameter at breast height (*D*) greater than 10 cm,
**2.1.2 Savanna (SA)** plots may be considered as natural vegetation type from all three agro ecological zones (AEZ). Trees
were typically between 5 to 10 m high and with a density of 50 to 100 trees (*D* > 10 cm) per hectare.  Due to their open nature,
these savanna formations were typically with an abundant ground layer of grasses and herbs.
**2.1.3 Agricultural field plots (AF)** were selected are close as possible to the FI and SA plots and, from discussions with local
village inhabitants, had been exposed to at least 10 years of cultivation. In Burkina Faso, the cropland study sites were cotton
based or cereals based fields. In Ghana, the cropping areas were monocultures of maize. In Nigeria, they were maize or mixture
of maize/cassava or legumes.
**2.2  Soil sampling**
At each of the 11 locations, soil samples were collected from FI, SA and AF. The size of the sampling area was 0.16 ha which
was divided into four 20 x 20 m subplots for soil sampling. Within each subplot at least five samples were taken from 0-5 and
5-10 cm depth using undisturbed soil sampling auger (Eijkelkamp Agrisearch Equipment BV, Giesbeek, The Netherlands).
Samples were subsequently air-dried and stored for laboratory analysis.





**2.2.1 Soil aggregate stability**
The wet seiveing method (Mathieu and Pieltain, 1998) was used to determine soil aggregate stability. This method consists of
passing air-dried soil samples through 4000 μm, 500 μm, 250 μm and 53 μm sizes sieves (not sequentially) to obtain three
aggregate fractions defined as "macroaggregates" (4 mm-500 μm), "mesoaggregates" (500-250 μm) and "microaggregates"
(250-53 μm). To obtain each aggregate class, 3 g of soil sample previously moistened by spraying with distilled water was
placed on sieves of either 500 μm (macroaggregates), 250 μm (mesoaggregates) or 53 μm (microaggregates). The sieves were
then placed on the wet sieving equipment, and shaken slowly backwards and forward for one hour until all the unstable
aggregates passed through the sieve mesh.
At the end of the sieving procedure, aggregate fractions were collected in a cup, oven dried at 105 °C for 24 hours and
then weighed. The sand fraction of each aggregate fraction was then determined after destruction of organic matter by adding
3 ml of hydrogen peroxide by heating till all bubbles disappeared from the soil-water mixture, after which the solution was
made up to 75 ml with distilled water and the soil particles dispersed using sodium hexametaphosphate. Afterwards, samples
were washed on a 0.5 mm sieve and then dried and weighed. The fraction of soil stable aggregates ($\Phi_A$) was then calculated
using the following formula (Bloin et al., 1990)
$$\Phi_A \; = \; (P_{ag} - P_s)/(P_e - P_s) \qquad\qquad (1)$$
where $P_{ag}$ = the dried total soil remaining in the sieve, $P_e$ = the weight of soil sample used and $P_s$ = weight of the sand in the
sample.
2.2.2 Particle size analysis
The separation of the sand, silt and clay fractions were done using Robinson-Köhn method. This method consists of destruction
of organic matter by hydrogen peroxide followed by particle dispersion with sodium hexametaphosphate, with subsequent
separation of silt and clay particles by sedimentation with sands by sieving (Mathieu and Pieltain, 1998).
2.2.3 Chemical analysis
Soil pH was measured using the electrode method in a ratio of soil / water of 1: 2.5. Total soil carbon content was determined
in an automated elemental analyzer (Vario MACRO cube, Elementar Germany). Soil total and available Fe were determined
by direct colorimetry after etching with concentrated hydrochloric acid and sodium hydrosulfite (Mehrotra, 1992).
2.2.4 Statistical analysis
In order to evaluate the potential joint effects of mean annual precipitation (v), land-use (L) and sampling depth (d) on the
three aggregate fractions, we fitted a mixed effect model allowing for stratified nature of the sampling design according to
$$\log_{10}\left[\arcsin(f_{dcp})\right] = \alpha_{000} + \alpha_{001}P_{A00p} + \gamma_i L_{00p} + \gamma_j d_{0cp} + U_{00p} + V_{0cp} + R_{dcp} \qquad , \qquad (2)$$



where $f_{dcp}$ is the aggregate fraction f as measured at depth d of core c in plot p; $\alpha_{000}$ is the overall mean value of f at 0 to 5 cm
depth for agricultural fields (AF) across the dataset (intercept term with all model input centered on the dataset mean annual
precipitation ($P_A$) of 1.01 m a$^{-1}$), $\alpha_{001}$ is a fitted variable describing the response of f to $P_A$, $\gamma_i$ is the response of f to the land use
indicator variable L (for which AF = 0, forest island (FI) = 1 and savanna (SA) = 2 ); $\gamma_j$ is the difference in f between the
upper and lower sampling depths for core c within plot p; $U_{00p}$ represents the variance associated with plot location (i.e. the
systematic component of the plot variation that is not accounted for by the precipitation and land use terms); $V_{0cp}$ is the within-
plot variation (i.e. the variance associated with the sampling of replicate cores within individual plots) and $R_{dcp}$ is the residual
variance.
In terms of the fixed components, it is worth noting that (2) can also be written as (ignoring subscripts where possible for
convenience)

$$f = \sin\left(10^{[\alpha_0 + \alpha_1 P_A + \gamma_i L + \gamma_j d]}\right) = \sin\left(10^{[\alpha_0 + P_A]} 10^{\gamma_i L} 10^{\gamma_j d}\right) \qquad , \qquad (3)$$

which illustrates the essentially multiplicative nature of the untransformed model. In terms of precipitation sensitivities,
Equation 3 may also be differentiated as (taking the indicator variables $\gamma_0$ and $\gamma_i$ as zero (= AF) for simplicity)

$$\frac{df}{d\langle P_A \rangle} = \alpha_1 \cdot \cos\left(10^{[\alpha_0 + \alpha_1 P_A]}\right) \cdot 10^{[\alpha_0 + \alpha_1 P_A]} \cdot \log(10) \qquad , \qquad (4)$$

Note that for the fitting of the mixed model, the input precipitations were centered on the dataset mean of 1.01 m a$^{-1}$. This
means that, once appropriately back transformed, the fitted intercept gives an estimate of f at the dataset mean precipitation
rather than the (relatively meaningless) $P_A$= 0 m a$^{-1}$.
**3. Results**
**3.1. Effects of rainfall pattern and land use on aggregate fractions**
Figure 2 shows the variations in the three aggregate fractions with land use type and precipitation (0 to 5 cm depth only)
with the fitted lines coming from the mixed model analysis of Table 2. For the micro aggregates (Fig 2a), there was a strong
increase in relative abundance with precipitation ($p < 0.001$) but no effect of land use ($p > 0.1$) with the intercept of -0.030
equating to a predicted $f_{micro}$ of $\sin(10^{-0.03}) = 0.803$ for agricultural fields (AF) at the dataset mean of 1.01 a$^{-1}$, and with the
associated coefficient of $0.976 \pm 0.272$ m$^{-1}$ equating to an increase of $0.975 \times [10^{-0.03} \cos(10^{-0.03})] \times \log(10) = 1.24$ m$^{-1}$, *viz.* with
each 10 mm increase in $P_A$ being associated with a relative increase in $f_{micro}$ of $1.24/0.803 = 1.6\%$. Although the fitted equation
is linear in form, due to the dual logarithmic and arcsine transformations, $f_{micro}$ is clearly a saturating function of. For example,
at a lower =0.80 a$^{-1}$ then $f_{micro} = \sin(10^{[-0.03 + (0.976 \times -0.201)]}) = 0.561$ and with the relative increase in $f_{micro}$ per 10 mm of $P_A$ equal
to 1.9%. Likewise, for the higher $P_A$ =1.20 a$^{-1}$ we obtain through equivalent calculations a predicted $f_{micro}$ of 0.994 and with
each 10 mm increase in rainfall being associated with an relative increase in $f_{micro}$ of just 0.2%. Although for the sake of clarity
(not shown in Fig 2a), from Table 2, it is also evident that there is an effect of depth ($p < 0.05$) with the regression coefficient


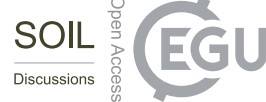

of -0.086 ± 0.029 m$^{-1}$ suggesting that $f_{micro}$ were typically 13.7% lower at 5 to10 cm depth than was the case for the upper 0 to
5 cm at the data set average of $P_A$ = 1.01 m a$^{-1}$. Due to the dual $log_{10} \times arcsine$ transformation employed as part of Equation 2,
there is a slight dependency of this (relative) depth difference on $P_A$ in the model with the lower layer modelled to be 13.1%
lower at $\langle P_A \rangle$ = 0.8 m a$^{-1}$ and 14.2% lower at $P_A$ = 1.20 m a$^{-1}$.

For both the mesoaggregates (Fig. 2b) and macroaggregates (Fig. 2c), very different patterns of variation were observed

with there being no dependence of aggregate fraction on $P_A$ but with effects of land-use being observed in both cases (Table
2). For example, again calculating at the data set average $P_A$ = 1.01 m a$^{-1}$ we obtain for estimates for $f_{meso}$ = sin($10^{-0.805}$) = 0.15
for AF and with forest island (FI) and savanna (SA) modelled to have $f_{meso}$ that were, on average, 122% and 67% higher
respectively – but with only the FI-AF difference being significant at $p$ < 0.05. As for $f_{micro}$ there was an effect of sampling
depth on $f_{meso}$ with values of the 5-10 cm depth typically being $10^{-0.141}$ = 26% lower than is observed at 0 to 10 cm depth.
Overall, the patterns observed for $f_{macro}$ were as for $f_{meso}$ (Fig 2c), but with the effect of sampling depth being a little less marked
(Table 2).

Also of interest in Table 2 are the variances associated with the random components, for which it can be seen that,

although for the microaggregates the between-plot variance ($\tau^2$) was slightly less than the residual variance ($\sigma^2$), for both the
meso and macroaggregates ($\tau^2 \gg \sigma^2$) indicating that there was much more systematic between-plot variation that could not be
accounted for by the either precipitation or land-use for the two larger aggregate types. For all three aggregate sizes examined,
the within-plot variance was the smallest component: This indicates that, after accounting for systematic land-use and
precipitation effects, that the variation within a plot was typically less than was between plots, and with this within-plot
variance also being typically less than the variation within individual soil cores after accounting for systematic depth effects.
There were higher ($p$ <0.05) proportion of stable meso and macro- aggregates in forest islands and natural savanna compared
to agricultural soils (Table 3).

### 3.2 Underlying basis of differences in aggregate fractions

Using Kendall's $\tau$ and taking mean values per plot (upper 0 to 5cm depth only), Table 4 details the strength of associations
between the three aggregate fractions as well as correlations with and between measures of soil citrate-, dithionate- and
pyrophosphate-extractable aluminium and iron, soil carbon and mean annual precipitation. This shows, as might be expected
from Fig. 2a, that for $f_{micro}$ there was a strong positive association with $P_A$ ($\tau$ = 0.50; $p$ < 0.0001), and with a weaker negative
association with pyrophosphate-extractable aluminium also of note ($\tau$ = -0.26; $p$ = 0.051). On the other hand, for $f_{meso}$ it was
the dithionate-extractable aluminium [Al$_o$] that showed the strongest (negative) correlation ($\tau$ = -0.28; $p$ = 0.032), and with
both dithionate-extractable iron [Fe$_d$] ($\tau$ = -0.26; $p$ = 0.068) and dithionate-extractable aluminium [Al$_d$] ($\tau$ = -0.26; $p$ = 0.072)
as well as soil [C] also being positively associated ($\tau$ = 0.26; $p$ = 0.047). Overall, across sites, there was a very strong association
between $f_{meso}$ and $f_{macro}$ ($p$ < 0.0001), with soil [C] appearing to be a much stronger determinant of the latter ($\tau$ = 0.42; $p$ =
0.0012). Also of note, [Fe$_d$] also showed a modestly strong correlation with $f_{macro}$ ($\tau$ = -0.25; $p$ = 0.053).



In order to separate out the potentially causative versus correlative factors, partial Kendall correlation coefficients $\tau_P$ were
subsequently employed. For example, for $f_{meso}$ – testing for [Al$_o$], [Al$_d$], [Fe$_d$] and [C] separately (whilst in each case
controlling for variation in the other three covariates) – all of [Al$_o$], [Al$_d$] and [Fe$_d$] were all found to be with $\left|\tau_P\right| < 0.22$
and with $p > 0.1$; the best of the four tested predictors being [C] for which $\tau_P = 0.23$ and $p = 0.093$. Although this result for
$f_{meso}$ must be regarded as negative, a similar analysis confirmed a unequivocal strong role for [C] in accounting for site-to-site
variations in $f_{macro}$ ($\tau_P = 0.39$; $p = 0.004$), although with all three other tested variables all having $\left|\tau_P\right| < 0.2$ and with an
associated $p > 0.2$. For $f_{micro}$ the same partial Kendall's analysis suggested nothing other than a strong role for $P_A$ in accounting
for the variations observed as already indicated (Tables 2 and 3). With the $f_{micro}$ vs. $P_A$ association already shown in Fig 2a,
Fig. 3 shows the nature of the significant $f_{macro}$ vs. [C] association across sites.
**4. Discussion**
Our data showed strong influence of precipitation on soil micro-aggregates whereas land use type influenced the larger
aggregate groups – meso and macro (Table 2). The gradual increase in stable soil micro aggregates ($f_{micro}$) with precipitation
may be a result of seasonal variation in soil moisture and soil drying-wetting cycles which has impact on soil microbial activity
often considered a binding agent in soil aggregate formations. Micro-aggregates may initially form by the progressive bonding
of primary particles of clay, SOM (soil organic matter) and cations, with fungal and bacterial debris giving rise to extremely
stable micro-aggregates (Bongiovanni and Lobartini, 2006; Bouajila and Gallali, 2008).
Macro-aggregates fall apart in response to major rainfall events due to disruptive forces (wetting and drop impact) which
contributes to release of more micro-aggregates during rainfall (Bach and Hofmockel (2015). It has, for example, been reported
that increasing soil moisture results in a lower shear strength of wet aggregates and consequently a higher vulnerability to
raindrop impact. Regardless of the aggregate hierarchy theory, drying/wetting plays a key role on macro turnover releasing
micro-aggregates (Tisdall et al., 1982; Six et al., 2004; Bach and Hofmockel, 2015) which may increase the local concentration
of enzymes to stimulate microbial activity and increase continual carbon turnover.
The fact that land use influenced meso and macro aggregates across locations is attributable to management benefits arising
from differences in soil organic carbon content and vegetation characteristics, explaining to some extent the positive
correlations observed between soil organic matter content and aggregate stability (Table 4 & Figure 2). Soil organic carbon is
known to improve aggregate stability via different mechanisms and by its different fractions as a result of inner sphere
interaction between the carboxyl groups and cations of the mineral structure through ligand exchange mechanism (Mikutta et
al., 2011). Although other organo-mineral interactions have also been proposed viz. hydrophobic interactions, cation bridges;
cation and anion exchange; and Van der Waals interactions, among others (Hanke et al., 2015, Hanke and Dick, 2017), these
have not been well investigated.
The higher proportion of macro-aggregates in forest islands and natural savanna than in the cultivated soils (Table 3) indicated
negative effects of cultivation on soil aggregation. In cropland, disaggregation of macro-aggregates due to frequent tillage
(Ouattara, 2007; Six et al., 2000) is known to be a key factor leading to less stable aggregates. This is because frequent plowing
leads to physical disruption of aggregates which is highly vulnerable to soil stability (Six et al., 2004). Moreover, plowing




causes loss of soil organic matter via increased mineralization with negative implications on aggregate stability. Similar results
have also been reported by Cerdà, (2000) who found higher soil aggregate stability in forest than in cropland in southern
Bolivia. Likewise Erktan et al. (2015) and Wang et al. (2012) reported decline in soil aggregate stability resulting from the
conversion of forest into crop land whilst Duchicela et al. (2013) and Zombre (2003) observed a decrease in aggregate stability
in cropland after decline in  vegetation cover exposing soils to crusting or compaction. Accumulation of organic matter through
litter decomposition, roots dynamics and soil biological activities (Bronick and Lal, 2005; Le Bissonnais et al., 2017) could
also account for the higher meso and macro aggregates of the forest islands and savanna than croplands. Bronick and Lal.
(2005) and Le Bissonnais et al. (2017) showed that roots act either by emeshment or by decompaction of the soil or by root
exudations, which bind soil particles and increase cohesion. Organic carbon is a major binding agent of aggregates (Mentler
et al., 2010).
The role of vegetation in forest land on macro-aggregates stability has also been attributed to diversity and species richness,
which is associated with functional diversity (Pagliai et al., 2004; Six et al., 2004; Ouattara et al., 2008; Gould, 2016).  Indeed,
vegetation cover may moderate the impact of drying-wetting (Bronick and Lal, 2005) with the litter protecting the soil from
the splash effect of the rains and the phenomena of suddent drying-wetting of the soil (Le Bissonnais et al., 2017). The roots
increase the magnitude of the drying-wetting cycle, promoting the structural stability of the soil. This may be one further
reason for the higher meso and macro-aggregates observed in the forest islands and savanna than crop lands (Table 3).
Our results showed significant correlation between soil properties and aggregates and this was confirmed by the very strong
association between $f_{\mathrm{meso}}$ and $f_{\mathrm{macro}}$ ($p < 0.0001$). It showed (Figure 2 and Table 2), confirming that accumulation of organic
carbon can improve aggregate stability and the soil's resilience to erosive forces. Positive relationships between iron oxides
content and soil stability have also been reported under cotton cropping systems in the Sudan zone of Burkina Faso (Ouattara,
2007; Ouattara, 2008).  Iron oxides are key components of  clay minerals (Six et al., 2004) as they serve as flocculants, binding
fine particles to organic molecules (Borggaard, 1983) with improved effects on aggregation. Römkens and Lindbo (1998)
showed that aggregation in soils was enhanced from combination of organic material, iron-aluminium oxides and clay
minerals.



**5. Conclusions**
Soil micro aggregate stability was not affected by land-use type but did systematically increase with greater annual
precipitation in West Africa whereas the larger fractions were influenced directly by land use type, being systematically lower
in agricultural soils than either natural savanna or in forest islands.  Soil organic carbon content and iron oxides were key
determinants of aggregates stability in the region. Contrary to our original hypothesis, these were, however, no differences in



aggregate stability between FI and SA. This suggests that other soil physical and chemical factors must underlie the West
African forest island phenomenon.
**6. Acknowledgments:** The authors are grateful to the Royal Society-DFID for funding the study through the Soil of Forest
Island in Africa (SOFIIA) ACBI project. We also thank the local village occupants for their collaboration and field
assistance.

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





**Table 1:** Details of study sites including land use type (cropland = 0, forest island =1, natural savanna = 2), geographical
coordinates, mean daily temperature of the coldest month ($T_{min}$), mean daily temperature of the hottest month ($T_{max}$), mean
annual precipitation ($P_A$) and WRB soil classification.

| Sites | Land use | Lat | Long | $T_{min}$ (°C) | $T_{max}$ (°C) | $P_A$(m) | Soil types |
|---|---|---|---|---|---|---|---|
| Koupela (KPL) (Burkina Faso) | 0 | 11.95157 | -2.40529 | 16.2 | 38.8 | 0.81 | Lixisol (Arenic, Rhodic) |
| | 1 | 11.95051 | -2.40536 | 16.2 | 38.8 | 0.81 | Lixisol (Arenic, Rhodic) |
| | 2 | 12.09921 | -2.25859 | 15.8 | 38.9 | 080 | Eutric Plinthosol (Lixic, Loamic) |
| Toece (TOE) (Burkina Faso | 0 | 11.82644 | -1.22018 | 17.3 | 38.2 | 0.83 | Lixisol (Arenic, Rhodic) |
| | 1 | 11.82578 | -1.22142 | 17.3 | 38.2 | 0.83 | Lixisol (Arenic, Rhodic) |
| | 2 | 11.74883 | -1.21682 | 17.3 | 38.2 | 0.83 | Stagnic Pisoplithic Plinthosol (Lixic, Loamic) |
| Hounde (HOU) (Burkina Faso) | 0 | 11.52748 | -3.54269 | 17.0 | 38.0 | 0.91 | |
| | 1 | 11.52774 | -3.54222 | 17.0 | 38.0 | 0.91 | Ferric Lixisol |
| | 2 | 11.32041 | -3.26029 | 17.7 | 37.8 | 0.95 | Stagnic Lixisols (Loamic, Hypereutric) |
| Kadomba (KAD) (Burkina Faso) | 0 | 11.49749 | -3.99781 | 16.4 | 37.7 | 0.95 | Stagnic Lixisols (Loamic, Hypereutric) |
| | 1 | 11.4987 | -3.9979 | 16.4 | 37.7 | 0.95 | Stagnic Lixisols (Loamic, Hypereutric) |
| | 2 | 11.74883 | -4.21682 | 15.1 | 38.0 | 0.91 | Stagnic Lixisols (Loamic Hypereutric) |
| Navrongo (NAG) (Ghana) | 0 | 10.86427 | -1.08127 | 18.9 | 38.4 | 0.91 | Stagnic Pisoplinthic Plinthosol (Lixic, Clayic) |
| | 1 | 10.86466 | -1.08091 | 18.9 | 38.4 | 0.91 | Stagnic Pisoplinthic Plinthosol (Lixic, Clayic, Humic) |
| | 2 | 10.78512 | -1.21984 | 19.0 | 38.2 | 0.98 | Stagnic Petric Plinthosol (Eutric,Arenic) |
| Changnaayili (CHN) (Ghana) | 0 | 9.37016 | -0.70318 | 20.1 | 37.4 | 1.10 | Pisoplinthic Plinthosol (Loamic, Ochric) |
| | 1 | 9.37222 | -0.70375 | 20.1 | 37.4 | 1.10 | Pisoplinthic Plinthosol (Abruptic, Loamic) |
| | 2 | 9.39866 | -0.59398 | 19.9 | 37.3 | 1.12 | Stagnic Petric Plinthosol (Eutric,Arenic) |
| Nkoranza (NKZ) (Ghana) | 0 | 7.5354 | -1.70812 | 19.5 | 33.6 | 1.27 | Abruptic Chromic Lixisol (Loamic, Cutanic, Profondic) |
| | 1 | 7.56341 | -1.71302 | 19.5 | 33.6 | 1.27 | Abruptic Chromic Lixisol (Loamic, Cutanic, Profondic) |
| | 2 | 7.65579 | -1.64400 | 20.1 | 34.6 | 1.24 | Abruptic Chromic Lixisol (Loamic, Cutanic, Profondic) |
| Wasim Okuta (WSM) (Nigeria) | 0 | 7.53256 | 2.76823 | 20.8 | 35.4 | 1.12 | Eutric petroplinthic Cambisol |
| | 1 | 7.52827 | 2.76886 | 20.8 | 35.4 | 1.12 | Eutric Arenosol (Humic) |
| | 2 | 7.52708 | 2.76785 | 20.8 | 35.4 | 1.12 | Rhodiv Luvisol (Arenic) |
| Ilua (ILU) (Nigeria) | 0 | 8.0045 | 3.40821 | 19.2 | 34.8 | 1.16 | Plinthosol (Arenic, Eutric) |
| | 1 | 8.00307 | -3.40896 | 20.0 | 35.0 | 1.07 | Rhodic Luvisol (Clayic) |
| | 2 | 7.9994 | 3.44503 | 20.0 | 35.0 | 1.15 | Ferric Lixisol |
| Onikpataku (ONP) (Nigeria) | 0 | 7.39044 | 3.02113 | 21.4 | 35.0 | 1.13 | Lixisol (Arenic, Rhodic) |
| | 1 | 7.38982 | 3.02017 | 21.4 | 35.0 | 1.13 | Plinthosl (Lixic) |
| | 2 | 7.39691 | 3.02048 | 21.4 | 35.0 | 1.13 | Plinthosol (Clayic, Eutric) |
| Elewere (ELE) (Nigeria) | 0 | 8.03883 | 3.44167 | 19.2 | 34.8 | 1.16 | Plinthosol (Arenic) |
| | 1 | 8.041 | 3.44171 | 19.2 | 34.8 | 1.16 | Rhodic Luvisol (Clayic) |
| | 2 | 8.0425 | 3.44224 | 19.2 | 34.8 | 1.16 | Eutric Cambisol (Arenic) |









Table 2: Estimates for linear mixed effects models relating variation in log × arcsine transformed aggregate fractions to precipitation and land-use type. For this analysis Mean Annual Precipitation $P_A$ estimates for each site have been centred on the dataset mean value of 1.01 m a$^{-1}$.

| Fixed effect | Microaggregates $R_m^2 = 0.17,\ R_c^2 = 0.59$ | | | Mesoaggregates $R_m^2 = 0.14,\ R_c^2 = 0.82$ | | | Macroaggregates $R_m^2 = 0.14,\ R_c^2 = 0.82$ | | |
|---|---|---|---|---|---|---|---|---|---|
| | Coef. | S.E | t | Coef. | S.E | t | Coef. | S.E | t |
| Intercept (Agricultural field) | -0.030 | 0.0036 | -0.82 | -0.805 | 0.101 | -7.94 | -0.990 | 0.127 | -7.82 |
| $P_A$(m) | 0.976 | 0.272 | 3.58 | 0.180 | 0.418 | 0.43 | 0.467 | 0.522 | 0.89 |
| Forest island | 0.007 | 0.093 | 0.07 | 0.354 | 0.141 | 2.50 | 0.383 | 0.177 | 2.17 |
| Savanna | -0.003 | 0.095 | -0.04 | 0.227 | 0.142 | 1.60 | 0.401 | 0.177 | 2.27 |
| Sampling depth | -0.086 | 0.029 | -2.97 | -0.141 | 0.024 | -5.90 | -0.106 | 0.029 | -3.62 |
| Random Component | Parameter | | | Parameter | | | Parameter | | |
| Within plot variance | 0.0097 | | | 0.0190 | | | 0.0177 | | |
| Between plot variance | 0.0387 | | | 0.1086 | | | 0.1735 | | |
| Residual variance | 0.0474 | | | 0.0337 | | | 0.0528 | | |

Table 3: Effect of land use on aggregates

| Aggregates (%) Land use | Macro aggregates | Meso aggregates | Micro aggregates |
|---|---|---|---|
| Cropland | 15.9±2.4 [b] | 17.8± 2.1[b] | 73.6±1.9[a] |
| Forest island | 32.3 ±2.2 a | 35.8± 1.9[a] | 73.5±1.8[a] |
| *Savanna | 32.0 ±2.1 [a] | 31.0 ± 1.8[a] | 74.3 ±1.9[a] |
| *Probability value* | 0.00*** | 0.000*** | 0.9ns |

DF : Degree of Freedom, SS : Square Sums, Ms : Means of Square, Pr : F Probability
Significant differences : * P=0.05 ; ** P=0.01 ; *** P=<0.001  ns= not significant





Table 4. Strength of association between the studied covariates as estimated by Kendall's τ (soil data for the 0 to 5 cm depth
only). Symbols used: $f_{micro}$ = micoaggregate fraction, $f_{meso}$ = mesoaggregate fraction, $f_{macro}$ = macroaggregate fraction, [$Fe_o$] =
oxalate extractable iron concentration, [$Al_o$] oxalate extractable aluminium concentration, [$Fe_d$] = dithionite extractable iron
concentration, [$Al_d$] = dithionite extractable aluminium concentration, [$Fe_c$] = pyrophosphate extractable iron concentration,
[$Al_c$] pyrophosphate extractable aluminium concentration, [C] = soil carbon concentration, $P_A$ = mean annual precipitation.
Relationships significant at p < 0.01 are shown in bold (with grey background) with those for which $0.01 \leq p \lesssim 0.05$ are
shown in italics.

| | $f_{micro}$ | $f_{meso}$ | $f_{macro}$ | [$Fe_o$] | [$Al_o$] | [$Fe_d$] | [$Al_d$] | [$Fe_c$] | [$Al_c$] | [C] |
|---|---|---|---|---|---|---|---|---|---|---|
| $f_{meso}$ | 0.21 | | | | | | | | | |
| $f_{macro}$ | 0.17 | **0.70** | | | | | | | | |
| [$Fe_o$] | -0.13 | 0.11 | 0.18 | | | | | | | |
| [$Al_o$] | -0.11 | *-0.24* | -0.16 | 0.23 | | | | | | |
| [$Fe_d$] | -0.16 | *0.24* | *0.25* | 0.32 | -0.30 | | | | | |
| [$Al_d$] | -0.16 | **-0.28** | -0.19 | 0.00 | **0.70** | -0.33 | | | | |
| [$Fe_c$] | -0.17 | 0.21 | 0.19 | -0.03 | **-0.52** | 0.64 | -0.40 | | | |
| [$Al_c$] | *-0.26* | -0.17 | -0.15 | -0.28 | 0.19 | -0.22 | 0.49 | 0.00 | | |
| [C] | 0.00 | *0.26* | **0.42** | 0.19 | 0.01 | 0.18 | -0.02 | 0.07 | -0.05 | |
| $P_A$ | **0.50** | 0.18 | 0.06 | -0.13 | -0.19 | -0.18 | -0.23 | -0.13 | -0.22 | 0.03 |




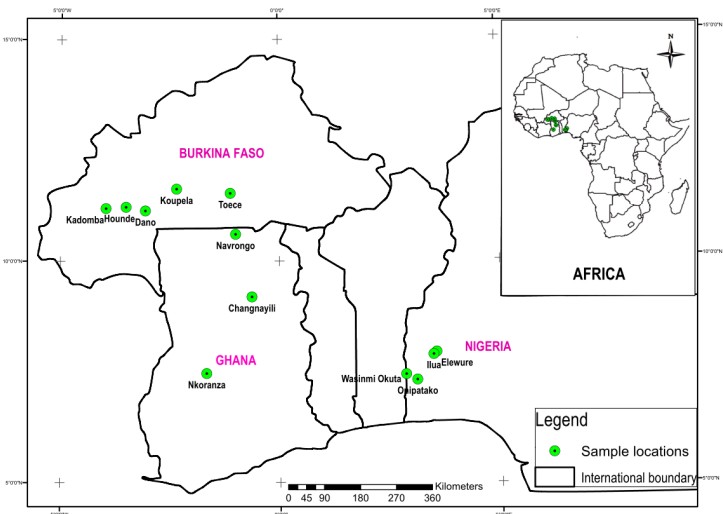

Figure 1: Location of study areas.
































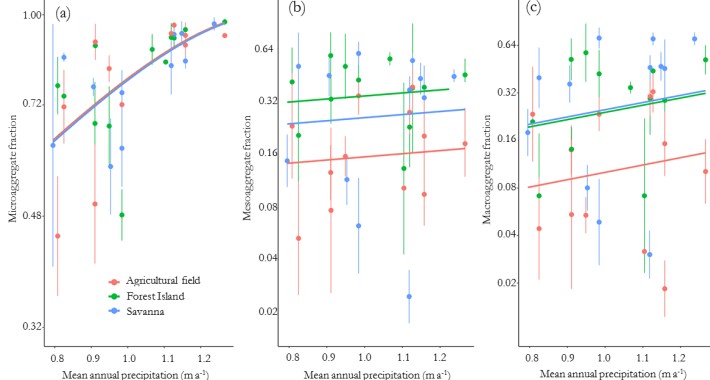


**Figure 2**. Effect of land-use and mean annual precipitation on 0 to 5 cm depth aggregate fractions. (a) microaggregates; (b)
mesoaggregates; (c) macroaggregates. Symbol and line colours as are indicated in panel (a), with the fitted lines representing the fixed
component of the model fits as summarised in Table 2.




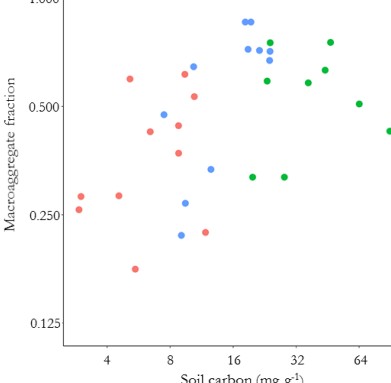


**Figure 3**. Relationship between soil carbon content and macro-aggregate fractions (0 to 5 cm depth). Symbols as in Figure 2.