# Peer review of "Soil Aggregate Stability of Forest Islands and Adjacent Ecosystems in West Africa"

_SOIL, 2019_

## Referee Comment (RC1) · Anonymous Referee #1 · 10 Mar 2020

In "Soil Aggregate Stability of Forest Islands and Adjacent Ecosystems in West Africa," Bougma et al. evaluate soil aggregate stability in forest islands, savannas, and agricultural fields at 11 locations representing a precipitation gradient in west Africa (Burkina Faso, Ghana, and Nigeria). Overall, they found that as mean annual precipitation increased, microaggregate stability also increased. However, they found no effects of land use on microaggregate stability. In contrast, precipitation had no effect on macro or meso aggregate stability. Land use was an important predictor of aggregate stability for these size classes, but there were no differences in stability between forest islands and savannas; agricultural fields generally had lower stability meso and macro aggregates than the other two land use types. Additionally, soil carbon content was strongly correlated with macroaggregate stability and macroaggregate stability and aluminum

content were strongly correlated with meso aggregate stability.

Overall, this study is scientifically sound, and the topic is of interest to the readership of SOIL. However, there are critical details missing from the methods section, the motivation for the study should be elaborated on, and the implications of this study must be added. Finally, while the authors suggest that the soil properties they evaluate in this study may lead to the development of forest patches (lines 22-23; lines 231-232), it seems more likely that changes in soil properties are the result of human establishment of forest patches (lines 22, 45). I suggest that the authors re-phrase and state that their study focuses on the effects of the development of patches with relatively luxuriant vegetation on soil aggregation.

Specific line comments are as follows:

Line 31 The implications of this study should be highlighted at the end of the discussion.

The introduction is short, but comprehensive. However, given that the authors find that soil mineralogy, climate, and agricultural disturbance are more important drivers of soil aggregation than vegetation per se, I suggest elaborating on the two sentences found in lines 38-42. The reader would benefit from more background information on these factors driving aggregate formation and develop the tension between physical and chemical drivers vs. biological drivers of aggregate formation. Additionally, the potential importance of these factors for different aggregate fractions should be described.

Line 78 This method is not used to "obtain each aggregate class," but rather, to assess aggregate stability in the different classes.

Lines 83-90 It is not clear to me why or how the sand fraction was determined for the stable aggregates from each size class. Currently, the methods suggest aggregate classes were separated by dry sieving and stability of these different fractions were determined by wet sieving. This all makes perfect sense to me. However, as it currently

reads, it sounds like these stable fractions were then subjected to particle size distribution analyses (%sand, silt, and clay) and afterwards, these solutions were sieved through a 0.5mm sieve. For the meso and micro aggregates (all <0.5mm), nothing should remain on the sieve. Why would only the sand content be relevant for only the stable aggregate pool and not the whole pool of soil in a given size fraction? And how would this method work for the smaller aggregate pools?

Line 95 Please describe the aluminum analyses. What are the different aluminum and iron fraction indicative of?

Line 99 Describe how the R2 values (in Table 2) were derived from the mixed linear models. How were the P-values derived in Table 3? What were the post-hoc tests used to evaluate drivers of differences among the land use types? How were statistically different distinctions made using these statistics? For instance, in the results, the authors state that precipitation and depth were the only significant variables for microaggregate stability, while land use, but not precipitation, was an important predictor for macro and meso aggregate stability. How was this determined? Describe how the statistics in Table 4 were derived.

Line 101 Here and throughout the manuscript, be careful about using the phrase "aggregate fractions." To me, aggregate fractions refer to the relative proportions of aggregates in different size classes derived from dry sieving. I believe what the authors are referring to are the relative aggregate stability in each dry fraction. Be clear that the factors affecting aggregate stability in the different fractions are the focus.

Line 102 Why did aggregate fraction need to be arcsin and log transformed?

Line 128 Delete the word "of."

Line 174 What implications do these results have for erosion? This was a major motivation in the introduction, but not mentioned in the discussion. Additionally, why did different parameters drive stability of the different aggregate fractions? This should be

explicitly addressed in the discussion.

Lines 181-186 It's not clear how this phenomena might enhance microaggregate stability. It might increase relative microaggregate abundance, but this parameter is not evaluated in this manuscript.

Line 189-191 Organic matter concentrations can both enhance and be enhanced by macroaggregates. This duality should be addressed in the discussion.

Lines 229-230 Only for macro and meso aggregates, respectively.

---

## Referee Comment (RC2) · Anonymous Referee #2 · 29 Apr 2020

The scope the submitted manuscript 'Soil Aggregate Stability of Forest Islands and Adjacent Ecosystems in West Africa' is interesting as it has a great ecological significance. Introduction and research methodology is well described. However the manuscript significantly needs a critical attention especially the results-discussion section. Below are the key comments to authors.

a). Major Comments:

1. In current study, did soil depth affect aggregate stability across sampling sites (zones)? This important information regarding depth effect on key soil indices is missing in current results and discussion section. For instance, the effect of land-use and mean annual precipitation on aggregate fractions has been depicted for only topsoil (0-5 cm) but why there is no information for second sampling (5-10 cm) depth (Figure

2 and Figure 3)?

2. Despite of the fact, physical fractionation of bulk soil was performed in current study, why the interaction of Organic carbon with only soil macro-aggregate is given in Figure 3? What about micro & meso aggregates, envisaged in this study?

3. Besides land use, was there any influence of soil type on aggregate stability across sampled sites?

4. Why the (depth wise) values for organic carbon, total nitrogen content, C: N ratio, bulk density, Cation exchange capacity (CEC) and pH of (bulk and fractionated) soil of sampling sites are not provided? It would have been more interesting if the C, N content of envisaged micro, meso and macro aggregates were also presented to get clear understanding which fraction sequestered more C content versus bulk soil under different land use systems!

5. Fe content was measured in this study and also inferred as key determinant of aggregate stability among sampling sites (regions) but why its impact on aggregate stability with respect to land use is discussed scarcely in current manuscript?

6. At the end of this study, it is still unclear that what are the (ecological) implications if soil micro aggregate fraction remained unaffected despite of land use change but only soil macro aggregates were affected (Line 227-229)! Does it mean micro aggregates are resistant to land use change? What should be inferred from this phenomenon particularly in the context of climate change? Please also state which aggregate fraction was (ecologically) significant in this study?

b). Minor comments:

- Contrary to results section (stated content wise), why the discussion section lacks of respective captions rather discussed holistically?

- The role of soil organic matter and organic carbon in aggregate stability has been redundantly discussed i.e. Line 187-191 and again in Line 203-208. Please check this.

Line 27: 'Water sieving method' should be corrected as 'wet sieving method'

Line 66: Which type of cultivation i.e. tillage type (conventional, reduced etc.) was applied to agricultural (AF) plots and up to what depth? Because tillage intensity greatly affects structural stability of soil aggregates.

Line69: Please state the date of soil sampling for the respective sampling locations

Line 71 (& 66): Instead of stating 'at least', kindly state exact number of samples taken (Line 71) and duration of cultivation (Line 66).

Line 183: Please briefly elaborate what 'wet aggregates' means here!

Line 175-186: Please check the paragraph spacing

Line 209-215: Please check the paragraph spacing

Line 195-198: Please discuss current results especially of AF fields in the context of plowing intensity

Line 197: Please briefly elaborate the extent of 'frequent plowing'! Does it refers to tillage intensity here!

Line 200-203: It is proved, well understood that conversion of forests to arable lands affects micro-biochemical indices of soil then what this study particularly unveils new for us?

Line 227: Were the differences among envisaged land use systems non-significant? What presumably lead to lack of land use change impact on soil micro aggregate stability especially among FI and AF land use systems?

Line 227: Does the phenomena of 'systematically increase' here means 'exponential increase'? Please briefly elaborate this.

Table 1 has not been cited at all in current manuscript. To which section it belongs!

---

## Author Comment (AC1) · 6 Jun 2020

Before any response, We would like to thanks editorial board and the reviewer for the valuable and relevant comment. Comment 1Âń However, there are critical details missing from the methods section, the motivation for the study should be elaborated on, and the implications of this study must be added. Finally, while the authors suggest that the soil properties they evaluate in this study may lead to the development of forest patches (lines 22-23; lines 231-232), it seems more likely that changes in soil properties are the result of human establishment of forest patches (lines 22, 45). I suggest that the authors re-phrase and state that their study focuses on the effects of the development of patches with relatively luxuriant vegetation on soil aggregation.

In response to your comments on the lines 22-23; lines 231-232, we

We have now changed our sentence and state that "To better understand the effects on soil quality of the development of these patches with relatively luxuriant vegetation, our study focused on the effect on soil aggregation of forest islands, nearby areas and natural savanna vegetation across a precipitation transect in West Africa". . ... Specific line comments are as follows: Line 31 Âń The implications of this study should be highlighted at the end of the discussion. We addressed this by ending the discussion with the following: Soil organic carbon and iron oxides contents as the most important factors influencing aggregate stability in West African ecosystems. By increasing soil structural stability, forest island contributes to soil erosion reduction and the control of land degradation.

Âń The introduction is short, but comprehensive. However, given that the authors find that soil mineralogy, climate, and agricultural disturbance are more important drivers of soil aggregation than vegetation per se, I suggest elaborating on the two sentences found in lines 38-42. The reader would benefit from more background information on these factors driving aggregate formation and develop the tension between physical and chemical drivers vs. biological drivers of aggregate formation. Additionally, the potential importance of these factors for different aggregate fractions should be described. Âż

We agreed with this point and we took it into account in the revised version by adding the following:

Tillage mixes the soil surface layers and exposes soil aggregates to wet–dry cycles (Roose, 1981; Beare et al., 1994; Ouattara, 1994; Whalen et al., 2003). Organic matter turnover is thereby increased and then weakened aggregates stability (Hadas, 1990). Whalen et al. (2003) and Bronick & Lal (2005) found also that in the absence of tillage, the addition of compost can increase macroaggregation and rhizospheric aggregate stability. Gicheru et al. (2004) ; AHMAD et al, (2014) reported more stable

soil aggregates in conservation agriculture system (minimum or no-tillage) compared to conventional tillage.

Ouattara et al., (2008) found positive correlation between microaggregate stability and soil base cation contents in Luvisol, whereas this correlation was negative in Lixisol, indicating a difference in chemical bonding mechanisms between the two soils. Soil organic matter is a main element in the cohesion and hydrophobicity of soil aggregates, thereby sensitivity to slaking of soil, depending on the soil intrinsic physico-chemical components (Igwe and al, 2009). Thus, persistent soil organic matter (humified, or recalcitrant) is key in microaggregates (53-250 $\mu$m) water stability, labile organic matter and microorganisms (transient, temporary) sustain water stable macroaggregates (> 250$\mu$m) making the later more vulnerable to agricultural management. There is a threshold of organic carbon above /beyond which additional input do not result to further increases in soil aggregate water stability (Amzeketa, 1999; Rillig 2004). Hydrous oxides, characteristic agent especially in microaggregate stabilization in weathered soil and aggregates of clay-size particles are associated to well-crystallized Fe oxides and poorly crystallized (Oxalate-extractable Fe) were most strongly active of coarse soil mineral particle. Ouattara et al., (2017) showed that the inorganic soil constituents, clay and iron oxyhydroxides in their amorphous form were controlling respectively 53 % (P<0.001) and 40 % (P<0.001) the soil aggregation under fields and fallow lands. Âń Line 78 This method is not used to "obtain each aggregate class," but rather, to assess aggregate stability in the different classes. Âż The sentence as formulated in the manuscript need to be reworded in appropriated term. Read Âń To obtain each aggregate class stability, 3 g of soil sample previously moistened by spraying with distilled water was placed, accordingly, on sieves . . . . . . ..Âż

Âń Lines 83-90 It is not clear to me why or how the sand fraction was determined for the stable aggregates from each size class. Currently, the methods suggest aggregate classes were separated by dry sieving and stability of these different fractions were determined by wet sieving. This all makes perfect sense to me. However, as it currently

reads, it sounds like these stable fractions were then subjected to particle size distribution analyses (%sand, silt, and clay) and afterwards, these solutions were sieved through a 0.5 mm sieve. For the meso and micro aggregates (all <0.5mm), nothing should remain on the sieve. Why would only the sand content be relevant for only the stable aggregate pool and not the whole pool of soil in a given size fraction? And how would this method work for the smaller aggregate pools?

Thank you for underlining this point, and the reviewer understanding is correct. Nevertheless, the sand content is relevant for the whole pool of soil in a given size fraction. The difference in sand content can be a source of error when comparing the aggregate stability of different treatments because sand is always stable to water sieving. The determination of sand content is used to correct the result for better appreciation of the aggregates stability.

Line 95 Please describe the aluminum analyses. What are the different aluminum and iron fraction indicative of?

The description of aluminum analyses is taken into account and integrated in the chemical analysis . Different aluminum and iron fraction have specific active role in aggregates stabilization. According to particle size, Fe crystallized and poorly crystallized amorphous forms (Oxalate-extractable Fe), well-crystallized Fe oxides plays in stabilization of coarse soil mineral particles and aggregation of clay-size particles respectively. Âń Line 99 Describe how the R2 values (in Table 2) were derived from the mixed linear models. How were the P-values derived in Table 3? What were the post-hoc tests used to evaluate drivers of differences among the land use types? How were statistically different distinctions made using these statistics? For instance, in the results, the authors state that precipitation and depth were the only significant variables for microaggregate stability, while land use, but not precipitation, was an important predictor for macro and meso aggregate stability. How was this determined? Describe how the statistics in Table 4 were derived.Âż

From generalized linear mixed-effects models Shinichi Nakagawa and Holger Schielzeth (2010; 2013), have derived two easily interpretable values of R2. The first is called the marginal R2m and describes the proportion of variance explained by the fixed factor(s) alone. The second is the conditional R2c, which describes the proportion of variance explained by both the fixed and random factors. In Table 3 the P values derived from the ANOVA table using land use as factor. Means comparison was made using the test of Newman-Keuls In figure 2a the slopes of the fitted lines are sharp, indicating an increase of stable microaggregates as the precipitation increase (climate more and more wet). The three fitted lines related to the land uses are overlapping, indicating that there is not differences in microaggregates stability between land uses. For mesoaggregates (Figure 2b) and macroaggregates (Figure 2c) the fitted lines are separated indicating differences between land use. But the slopes of the lines are not sharp, indicating low influence of rainfall increase. Table 4 derived from Kendall's correlation. It is like Pearson correlation but it is used when data are not normally distributed.

Âń Line 101 Here and throughout the manuscript, be careful about using the phrase "aggregate fractions." To me, aggregate fractions refer to the relative proportions of aggregates in different size classes derived from dry sieving. I believe what the authors are referring to are the relative aggregate stability in each dry fraction. Be clear that the factors affecting aggregate stability in the different fractions are the focus.Âż

This is addressed using when appropriate "aggregate size fractions"

Âń Line 102 Why did aggregate fraction need to be arcsin and log transformed? Âż

The transformation was done because the data were not normally distributed

Line 128 Delete the word "of." It is done. Âń Line 174 What implications do these results have for erosion? This was a major motivation in the introduction, but not mentioned in the discussion. Additionally, why did different parameters drive stability of the different aggregate fractions? This should be explicitly addressed in the discussion.Âż

It is right. The abundance of soil water stable aggregates may prevent the susceptibility of the ecosystem to erodibility. Because of the hierarchical ordering of aggregates and their binding agents, microaggregate stability is higher and less dependent on agricultural management than macroaggregate stability (Lado et al., 2004; Six et al., 2004). Macroaggregation depends on temporary binding agents (fine roots, fungi hyphae, labile organic matter, etc.) and is considered to be sensitive to the changes in organic matter levels caused by ploughing. In contrast, microaggregates depends on strong binding agents (clay charges, colloids, ions , persistent organic matter, etc.) then show relatively high stability in response to physical disruption (Tisdall & Oades, 1982, Ouattara et al., 2008).

Lines 181-186 It's not clear how this phenomena might enhance microaggregate stability. It might increase relative microaggregate abundance, but this parameter is not evaluated in this manuscript. Line 189-191 Organic matter concentrations can both enhance and be enhanced by macroaggregates. This duality should be addressed in the discussion.

Alright, It increase relative microaggregate abundance. Also, right that soil aggregates can protect organic matter from rapid decomposition.

Lines 229-230 Only for macro and meso aggregates, respectively.

Please also note the supplement to this comment:
https://soil.copernicus.org/preprints/soil-2019-87/soil-2019-87-AC1-supplement.pdf

---

## Author Comment (AC2) · 6 Jun 2020

Before any response, We would like to thanks editorial board and the reviewer for the valuable and relevant comment.

Anonymous Referee #2 The scope the submitted manuscript 'Soil Aggregate Stability of Forest Islands and Adjacent Ecosystems in West Africa' is interesting as it has a great ecological significance. Introduction and research methodology is well described. However the manuscript significantly needs a critical attention especially the results-discussion section. Below are the key comments to authors. a). Major Comments: 1. In current study, did soil depth affect aggregate stability across sampling sites (zones)? This important information regarding depth effect on key soil indices is missing in cur-

rent results and discussion section. For instance, the effect of land-use and mean annual precipitation on aggregate fractions has been depicted for only topsoil (0-5 cm) but why there is no information for second sampling (5-10 cm) depth (Figure 2 and Figure 3)? 2. Despite of the fact, physical fractionation of bulk soil was performed in current study, why the interaction of Organic carbon with only soil macro-aggregate is given in Figure 3? What about micro & meso aggregates, envisaged in this study? We put emphasis on the soil top layer because soil erosion starts on soil surface. The soil surface is the one in contact with rain drops, yet the stability of soil aggregates is an indication of its susceptibility to water erosion. In addition, the difference between land uses was higher on the top layer than in the sub layer. We did not find any significant interaction between meso aggregates, micro aggregates and soil organic matter content. That is why we don't have figures for these relationships

3. Besides land use, was there any influence of soil type on aggregate stability across sampled sites?

Problably, but we are exploring it in another paper.

4. Why the (depth wise) values for organic carbon, total nitrogen content, C: N ratio, bulk density, Cation exchange capacity (CEC) and pH of (bulk and fractionated) soil of sampling sites are not provided? It would have been more interesting if the C, N content of envisaged micro, meso and macro aggregates were also presented to get clear understanding which fraction sequestered more C content versus bulk soil under different land use systems! That is right but, the carbon fractionation is part of another study, this is why data were not show here.

5. Fe content was measured in this study and also inferred as key determinant of aggregate stability among sampling sites (regions) but why its impact on aggregate stability with respect to land use is discussed scarcely in current manuscript?

IT was not easy because Fe content is mainly inferred to soil type, climate condition and soil hydrology. Soil concretions formation, aluminum and iron oxides formation are

rather pedologic processes than effect of land use.

6. At the end of this study, it is still unclear that what are the (ecological) implications if soil micro aggregate fraction remained unaffected despite of land use change but only soil macro aggregates were affected (Line 227-229)! Does it mean micro aggregates are resistant to land use change? What should be inferred from this phenomenon particularly in the context of climate change? Please also state which aggregate fraction was (ecologically) significant in this study?

Macroaggregation depends on temporary binding agents (fine roots, fungi hyphae, labile organic matter, etc.) and is considered to be sensitive to the changes in organic matter levels caused by ploughing. In contrast, microaggregates depends on strong binding agents (clay charges, colloids, ions, persistent organic matter, etc.) then show relatively high stability in response to physical disruption (Tisdall & Oades, 1982, Ouattara et al., 2008). There is a hierarchy order in aggregates formation; microaggregates binding gives mesoaggregates and their binding gives macroaggregates. Soil quality is best with the abundance of stable macroaggregates.

b). Minor comments: - Contrary to results section (stated content wise), why the discussion section lacks of respective captions rather discussed holistically? - The role of soil organic matter and organic carbon in aggregate stability has been redundantly discussed i.e. Line 187-191 and again in Line 203-208. Please check this.

It is just an option to have the discussion in its actual format. Thanks, we address the redundancy.

Line 27: 'Water sieving method' should be corrected as 'wet sieving method' We aggree with your observation. This is took account in the manuscript Line 66: Which type of cultivation i.e. tillage type (conventional, reduced etc.) was applied to agricultural (AF) plots and up to what depth? Because tillage intensity greatly affects structural stability of soil aggregates.

Cultivation is cropping, not tillage.

Line69: Please state the date of soil sampling for the respective sampling locations Line 71 (& 66): Instead of stating 'at least', kindly state exact number of samples taken (Line 71) and duration of cultivation (Line 66).

Line 183: Please briefly elaborate what 'wet aggregates' means here! Line 175-186: Please check the paragraph spacing Line 209-215: Please check the paragraph spacing

These remarks were taken into account.

Line 195-198: Please discuss current results especially of AF fields in the context of plowing intensity Line 197: Please briefly elaborate the extent of 'frequent plowing'! Does it refers to tillage intensity here! Line 200-203: It is proved, well understood that conversion of forests to arable lands affects micro-biochemical indices of soil then what this study particularly unveils new for us?

We did some changes in the statements

Line 227: Were the differences among envisaged land use systems non-significant? What presumably lead to lack of land use change impact on soil micro aggregate stability especially among FI and AF land use systems? Line 227: Does the phenomena of 'systematically increase' here means 'exponential increase'? Please briefly elaborate this. Table 1 has not been cited at all in current manuscript. To which section it belongs!

The conclusion has been revised and the Table 1 called in the text.

Please also note the supplement to this comment:
https://soil.copernicus.org/preprints/soil-2019-87/soil-2019-87-AC2-supplement.pdf